# First Impressions Matter: Immune Imprinting and Antibody Cross-Reactivity in Influenza and SARS-CoV-2

**DOI:** 10.3390/pathogens12020169

**Published:** 2023-01-21

**Authors:** Samantha M. King, Shane P. Bryan, Shannon P. Hilchey, Jiong Wang, Martin S. Zand

**Affiliations:** 1Department of Medicine, Division of Nephrology, University of Rochester Medical Center, Rochester, NY 14642, USA; 2Clinical and Translational Science Institute, University of Rochester Medical Center, Rochester, NY 14618, USA

**Keywords:** influenza, immune imprinting, SARS-CoV-2, antigenic distance, multidimensional assay, cross-reactivity, memory B cells

## Abstract

Many rigorous studies have shown that early childhood infections leave a lasting imprint on the immune system. The understanding of this phenomenon has expanded significantly since 1960, when Dr. Thomas Francis Jr first coined the term “original antigenic sin”, to account for all previous pathogen exposures, rather than only the first. Now more commonly referred to as “immune imprinting”, this effect most often focuses on how memory B-cell responses are shaped by prior antigen exposure, and the resultant antibodies produced after subsequent exposure to antigenically similar pathogens. Although imprinting was originally observed within the context of influenza viral infection, it has since been applied to the pandemic coronavirus SARS-CoV-2. To fully comprehend how imprinting affects the evolution of antibody responses, it is necessary to compare responses elicited by pathogenic strains that are both antigenically similar and dissimilar to strains encountered previously. To accomplish this, we must be able to measure the antigenic distance between strains, which can be easily accomplished using data from multidimensional immunological assays. The knowledge of imprinting, combined with antigenic distance measures, may allow for improvements in vaccine design and development for both influenza and SARS-CoV-2 viruses.

## 1. Introduction

The emergence of immunologic memory to viral pathogens is a fundamental feature of the adaptive immune system, resulting in more robust and rapid immune responses upon subsequent re-infection [1,2]. These responses are most often directed at viral surface antigens, for example, the influenza hemagglutinin [3] and the SARS-CoV-2 spike protein [4,5]. Each large antigen contains numerous potential binding sites for B- and T- cell receptors, termed epitopes (Figure 1A) [6,7]. Effective antibody responses to viral pathogens rely on B cells that produce IgG antibodies that bind to different epitopes in the antigen with critical functions (e.g., receptor binding domains, membrane fusion regions), and thus prevent or “neutralize” infection. Most antibody responses are a mixture of these neutralizing and non-neutralizing antibodies, the balance of which determines the effectiveness of the response.

Both circulating viral pathogens [8] and human immune systems [9,10] are constantly undergoing adaptation. Viral pathogens undergo mutations in their surface antigens. Those mutations in epitopes that allow escape from the binding of antibodies produced from prior exposure to related, but not identical, epitopes confer a selective advantage to new viral strains. Conversely, memory B cells (MBCs) that recognize epitopes similar but not identical to those previously encountered will rapidly become activated and produce memory B cells, antibodies, and long-lived plasma cells. Indeed, the expansion of such MBCs and antibodies from repeated exposure to homologous or identical epitopes is the basis for inducing protective immunity by seasonal vaccination to influenza viruses, and now SARS-CoV-2 [10].

The ability of a mixture of serum antibodies to bind viral antigens (e.g., influenza HA, SARS-CoV-2 spike protein) from more than one strain or variant has been referred to as cross-reactivity [11]. Antibody cross-reactivity can occur when two antigens from different viral strains share one or more conserved epitopes targeted by a single antibody clone (Figure 1B). Cross-reactivity also occurs when sera contains a mixture antibodies with different epitope specificities, and the two antigens from different strains each contain at least one of the targeted epitopes. Thus, cross-reactivity is a function of the mixture of specific antibodies, and the composition of epitope targets within the antigens.

Imprinting is the phenomenon where the first exposure to a viral antigen shapes the immune response to subsequent exposures to related antigens that have a mixture of shared previously encountered and new epitopes (Figure 1C). It is a term that describes the influence of immune memory on subsequent immune responses and has also been referred to as “Original antigenic sin” (OAS) [12], antigenic seniority [13], back boosting [14,15] or antigenic imprinting [16]. The phenomenon was first identified in 1960 in studies of antibody-mediated immune responses to H1N1 influenza infection and vaccination, where subjects generated the highest level antibodies against the first influenza virus strain encountered in early childhood after exposure later in life to otherantigenically similar strains [12,17,18].

Subsequent work has shown that this phenomenon occurs when the newer viral strain shares some epitopes with the strain of first exposure but also has other antigenically distinct epitopes. This regularly occurs in circulating viruses under selective evolutionary pressure from population immunity or multiple species host environments, where mutation regularly occurs altering some epitopes in viral proteins in subsequent, emerging strains. For the host, this can lead to an immune response heavily biased towards the shared and previously encountered epitopes, delaying or dampening responses to novel epitopes (e.g., receptor binding site mutations) on the newer viral strain. This phenomenon has been referred to by some as “deceptive imprinting” [19]. It is due to a mixed primary and secondary memory recall response to newly and previously encountered epitopes, respectively. Thus, immune imprinting has two faces: it is desirable when a response to shared epitopes results in viral neutralization, but may be detrimental when a response to a non-protective epitope dominates. Imprinting has been observed after infection with a variety of different viruses in mice [20], rabbits [21], ferrets [22], nonhuman primates [23], and humans [12].

In this article, we will focus on the conceptual evolution of immune imprinting in the immunology literature, specifically in regards to B cell and antibody anti-influenza virus responses, and the current understanding of the mechanism of cross-reactive antibody generation by memory B cells. Additionally, we will discuss current evidence showing imprinting of the SARS-CoV-2 spike protein (S) and potential implications for repeat COVID-19 vaccination.

## 2. History of the Concept

The concept of immune imprinting, initially termed “Original antigenic sin”, was described in 1960 by Thomas Francis Jr [12]. Over the course of several years of comprehensive study, Francis described that antibody responses elicited against the first influenza A virus (IAV) infection strain encountered in early life would continue to be the dominant antibody response as the subjects grow older and subsequent influenza infections occurred [17,18]. Later infections with similar antigenic strains would enhance the original antibody to maintain it at the highest levels throughout their life [24]. This suggested that the first IAV exposure highly influenced antibody repertoire formation and future immune responses.

Following it’s initial discovery, immune imprinting received little attention from the scientific community, despite the H3N2 pandemic of 1968 and the H1N1 pandemic of 1977. In 2012, OAS would come to be known as “antigenic seniority,” a term that more accurately describes the phenomenon and does not carry a negative biblical connotation. Lessler et al. developed the term antigenic seniority to encapsulate the heightened response to strains and viruses seen early in life, not just the first strain exposure [13]. Thus early exposure makes a strain “senior” with an imprinted immune response, whereas strains encountered later in life are deemed to be “junior”. The term indicates the relationship between a person’s age and their immune response to more recent and novel influenza strain [25]. However, was not until the 2009 H1N1 pandemic that immune imprinting re-emerged as an area of research interest [11,20,25,26,27].

Subsequent studies have demonstrated that individuals mount more robust responses to influenza strains encountered early in life, and that this robust response to novel epitopes peaks at around 7 years of age [13]. Others have noted that the immune system preferentially produces antibodies to previously encountered epitopes over antibodies with more neutralizing potential against new epitopes, resulting in a weakened immune response [28]. This phenomenon, also termed “antigenic interference,” considers the detrimental impact of preexisting antibodies resulting in a weaker secondary immune response [29,30]. Others have suggested the alternate term “antigenic interaction” [24], as the mechanism for this appears to involve competition within the germinal center reaction rather than direct cell-to-cell signaling between memory and naive B cells [31] discussed further in Section 4 below.

In 2016, the concept of influenza HA imprinting was highlighted in a comprehensive study that suggested birth year was highly associated with the infection rates of avian flu H5N1 and H7N3 [32]. Significantly, influenza strains circulating during an individual’s childhood correlated with protection against strains in the same phylogenetic group later in life, due primarily to conserved epitopes in the HA head domain [32]. Individuals born before 1968, when group 1 influenza circulation predominated, were less likely to be infected with the H5N1 virus, which contains a group 1 HA protein. Those born after 1968 during a period of group 2 influenza prevalence enjoyed protection from the H7N3 virus, which carries a group 2 HA protein. The study proposed a method for identifying which age groups are at risk of developing a serious infection during any given influenza epidemic [32]. This work suggested that strain exposure and immune history should be considered when predicting the effect an influenza epidemic will have on a population. Importantly, this work showed strong evidence that previous exposure history could contribute to protection from influenza strains within the same phylogenic group, which are mostly homologous in the stalk subdomain of HA [33].

## 3. Immune Imprinting and Influenza

The concept of immune imprinting is intertwined with several characteristics of influenza virus antigenic structure and seasonality. The major antigenic target for influenza is the surface hemagglutinin protein. The HA1 portion of the protein, often referred to as the globular head, contains the globular head region with the sialic acid binding site, critical for viral binding to host cells. The HA2 portion contains the fusion apparatus necessary for the fusion of the virus with the host cell and the subsequent injection of the viral genome. Both portions of the HA protein contain numerous epitopes that are targets of neutralizing or non-neutralizing antibodies. Mutations in the HA protein lead to different strain variants, and those mutations that can evade prior influenza immunity generally lead to dominant strains in any year. Such mutations are not always due just to amino acid changes. For example, the addition or deletion of N-glycosylation sites can shield or unmask HA epitopes and alter imprinting [34,35]. Because influenza circulates seasonally, and worldwide, such mutations occur often. Emergent seasonal strains thus contain HAs with a combination of conserved and mutated epitopes, setting up conditions for imprinting to influence immune responses to new strains. Thus, the imprinting immune response to influenza is a combination of primary (i.e., naive) and secondary (i.e., memory) B cell immune responses (Figure 2A).

Imprinting is determined by the cross-reactivity of antibodies encoded in a host’s memory B cells (MBCs) with the new strain. This cross-reactivity can be predicted by the phylogenetic distance between the previously encountered and new influenza strains [36]. When considering the phylogeny of HA proteins there are two main groups, group 1 and group 2. Group 1 includes H1 and H5 whereas group 2 has H3 and H7. These groupings are based on phylogenetic temporal emergence and sequence similarities [37]. When an immune system is primed with an HA from one group, there is a greater cross-reactive antibody response to the conserved regions of other HAs within that same group, also termed heterosubtypic cross-reactivity [38]. Most of this cross-reactivity is due to amino acid sequence similarities between the stalk regions of strains within the same group. This resemblance is what makes up imprinting as there is a higher level of response even upon the first exposure to another related strain (Figure 2B).

In addition to different strains’ subtypes, there is also antigenic drift within the same strain subtype. Marchi et al. focused on three of the past H1N1 strains recommended for inclusion in influenza vaccines, A/Brisbane/59/2007 which was used for the 2008/2009 and 2009/2010 seasons, A/California/07/2009 which was used for the flu seasons from 2010/2011 to 2016/2017, and A/Michigan/45/2015 which was used for the seasons 2017/2018 and 2018/2019 [39]. It shows that there are different levels of cross-reactivity and protection when looking at immune responses to these strains within the H1 subtype as measured by hemagglutination inhibition (HAI)and single radial hemolysis assays using human serum samples [39]. These high serum IgG binding cross-reactivity between the A/California/07/2009 and the A/Michigan/45/2015 H1N1 even when the California 2009 first began to circulate. Additionally, within the elderly adult population, there was a greater number of cross-reactive and cross-protective antibodies seen with all three strains likely due to their primed and repeated exposures to subtypes that circulated earlier in their lives [39]. This shows that imprinting involves more than different viral strains or groups subtypes and lineages, but can also have an impact within a strain subtype that has experienced antigenic drift. Thus one of the benefits of imprinting is that the immune system can quickly respond to different, yet related, viruses or multiple strains and subtypes of the same virus.

**Figure 2 pathogens-12-00169-f002:**
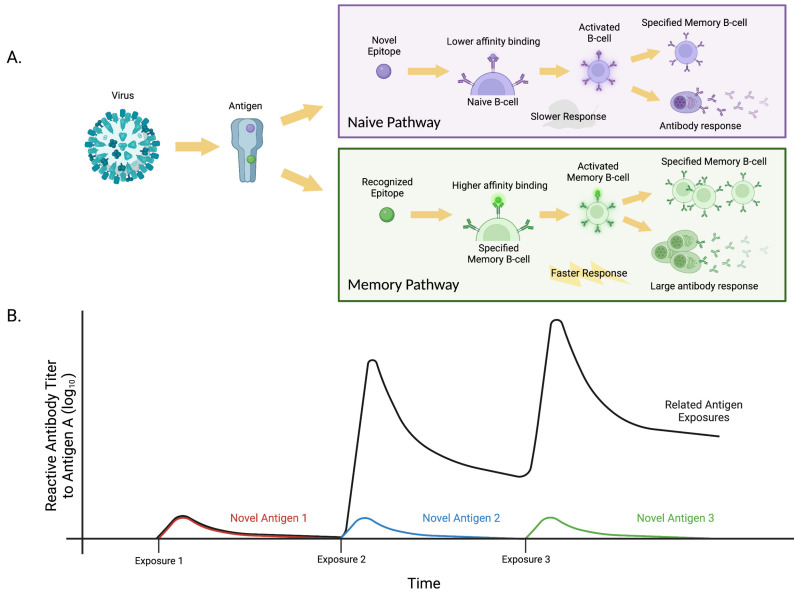
Variability in the speed and intensity of B cell immune responses as a result of imprinting. (**A**) B cells may follow one of two pathways when dealing with antigenic threats: the naive pathway or the memory pathway. Upon exposure to new epitopes, B cells will follow the naive, I pathway, responding to these epitopes with a lower binding affinity using IgM. Subsequent exposures to recognized epitopes will cause B cells to follow the memory pathway with a higher binding affinity using IgG. Memory responses to epitopes will result in the expansion of MBC pools, leading to a greater number of MBCs and plasma cells specific to that epitope [40]. These responses to recognized epitopes will be faster and larger due to the preexistence of MBC colonies that have already undergone selection and affinity maturation [41]. (**B**) B cell immune responses resulting from multiple exposures to related antigens with shared epitopes (black) or unique antigens with no shared epitopes (red, blue, green) showing the possible back boosting of preexisting antibodies associated with multiple exposures to conserved, previously recognized epitopes. The memory response to the related antigens, with recognized epitopes, is a faster, more robust response than that observed for novel antigens that follow the naive pathway, responding to new epitopes. Created with BioRender.com.

## 4. The Role of Memory B Cells in Imprinting

Imprinting is intimately related to mixed immune responses, those involving activation of both naive and memory B cells within the germinal center (GC) upon exposure to a new influenza antigen. Due to the back-boosting effect from imprinting, the MBC pool recognizing shared epitopes derived from antigens encountered earlier in life will enlarge and evolve upon exposure to a new, but related antigen that contains both shared and novel epitopes. The result will be the rapid expansion of the MBCs recognizing the shared epitopes, out-competing the naive response to the novel epitopes [3]. In further support of this concept, it has been shown that upon reactivation of MBCs targeting a conserved epitope, MBC colonies encoding cognate antibodies will grow in size. The more conserved these epitopes are across viral variants, the greater the clonal expansion [42]. For influenza responses, the size of MBC populations reactive to specific influenza HAs positively correlates with the level of antibodies produced that are reactive to that HA protein during infection, providing additional support for the idea that preferential MBC selection is a significant aspect of imprinted responses [43]. This clonal expansion of preexisting MBCs results in a strong, albeit short-lived response when compared to the clonal expansion of new MBCs produced after exposure to a foreign antigen. It was also found that new MBC colonies have a broader antibody repertoire and that these antibodies continue to increase in affinity several months after the antigenic insult along with the evolution of B cells [42], thereby increasing the imprinting effect prior to the next antigenic exposure. Indeed at the clonal level, the expansion and contraction of clonal populations within the GC is analogous to a Darwinian process, whereby B cells compete for resources, with higher affinity B cells outcompeting lower affinity ones [31].

These aspects of imprinting are, however, in some ways a double-edged sword. In many cases, the memory response to an antigen with moderate sequence homology to the originally encountered antigen can be less effective at viral neutralization than a naive response. In a primary immune response, naive B cells are activated followed by division, class switching and affinity maturation of B cell clones within the germinal center [44,45,46]. Dominant clones with high antigen affinity emerge via competition within the germinal center [44,47]. Alternatively, memory B cells responding to previously encountered epitopes are activated and expand more rapidly than the naive B cells responding to new epitopes, IgG secreting B cell clones recognizing new epitopes are less abundant. This results in lower concentrations of IgG against the newer variant epitopes seen by naive B cells, and paradoxically a less effective immune response to the non-identical virus [48]. Such ongoing mutation of circulating, seasonal strains of influenza are termed antigenic drift. This phenomenon implies that initial epitope exposure will create B-cell memory, and this imprinting impedes the production of efficient neutralizing antibodies after subsequent exposure to mutated epitopes. Such mutations occur under selective pressure in individuals with antibody-mediated immunity and are often present on the viral binding site for the host protein [8]. Finally, an imprinted B cell immune response to a newer viral surface protein can serve to facilitate, rather than block, viral infection through antibody dependent enhancement (ADE). ADE occurs when non-neutralizing antibodies bind to a pathogen’s surface epitopes and allow it to more easily infect host cells. This phenomenon has already been observed in cases of animal models for influenza [49], and is currently being studied in relation to SARS-CoV-2 [50,51].

It is known that vaccination or exposure to novel influenza virus strains can result in the selection and further consolidation of MBCs that target non-protective viral epitopes. Influenza viruses have been known to exploit this aspect of immune imprinting and prevent the immune system from effectively combating the virus [52]. When non-neutralizing HA epitopes are conserved across influenza strains, the virus is able to take advantage of a host’s immune imprinting response against other influenza HA proteins it encountered in the past containing the same non-neutralizing epitopes [28]. Widely conserved neutralizing epitopes on the HA head domain are rare because any individual exposed to a strain with this epitope would have some degree of immunity against every other influenza strain containing that epitope. This results in selective pressure to change neutralizing epitopes. While this is true for the head domain of HA proteins, the influenza HA stalk domain is unique. The stalk domain contains many widely conserved neutralizing stalk epitopes, which are important for viral fusion with target cells, and are hidden due to pre-fusion protein conformation. Thus, this portion of the HA protein is immunosubdominant and is seldom targeted by host immune systems due to its inaccessibility. Despite the inaccessibility of these conserved epitopes on the HA stalk, they have been a major target for universal vaccination strategies [34].

## 5. Measuring Imprinting and Antibody Cross-Reactivity

A variety of assays, combined with analytics, have been used over the years to measure B cell and antibody imprinting, including antibody binding assays, functional HA inhibition, and in vitro viral neutralization. They all take as a premise that four necessary conditions need to be demonstrated if imprinting has occurred: (1) antibodies and/or memory B cells generated against a viral protein from a prior immune encounter are present; (2) at least some homologous epitopes are shared by the imprinted viral protein, and a new viral protein subsequently encountered; (3) the newer viral protein has mutated epitopes not recognized by the imprinted memory B cells; (4) challenge (infection or vaccination) with the new, but related, viral protein triggers a B cell immune response dominated by the production of antibodies against the epitopes shared between the older and newer viral proteins. In a sense, one could describe imprinting as creating the conditions highly favorable for viral immune evasion, where the desired immune response is a production of neutralizing antibodies against the new viral epitopes that result in viral neutralization or clearance, but the actual response results in a predominance of antibodies against previously seen viral epitopes. Thus, measurement of B cell imprinting has several components: (1) measurement of antibody binding at the whole antigen level and, ideally, at the individual epitope level; (2) delineation of shared and discordant epitopes among the antigen variants; (3) functional assays to assess the effect of antibody binding on virulence and infectivity; (4) analytic methods to quantitatively describe the degree of immunologic similarity or discordance between related antigen variants (e.g., antigenic cartography) and host antibody responses (e.g., immune repertoire cartography).

### 5.1. Measuring Antibody Cross-Reactivity

The hemagglutinin inhibition (HAI) [53], enzyme-linked immunosorbent assay (ELISA), multiplex bead-based assays (e.g., mPLEX-Flu) [54], and optical reflectometry-based immobilized antigen and protein fragment assays [55], have all been used to measure the anti-HA antibody responses to various strains of influenza virus [56,57]. Current practice has shifted away from single dimensional assays (e.g., ELISA, HAI) as they are impractical for measuring antibody cross-reactivity against large numbers of antigen/sera combinations, which must be tested individually [58]. Multidimensional assays (MDAs) provide improved measures of IgG cross-reactivity against large panels of influenza HA or SARS-CoV-2 antigenic variants, as they can test antibody binding to dozens of antigens in a single sera sample. These high throughput assays decrease the time, sera, and labor needed to measure antibody binding against large panels of antigens without sacrificing sensitivity, and are therefore a practical means to measure antibody cross-reactivity while reducing batch variance [58].

One example of an MDA is the mPlex-Flu assay which utilizes a multiplex, bead-based, assay allowing for readings of fluorescent intensity to be measured [59]. This fluorescence can then be converted into concentrations of different immunoglobulins that bind to the selected antigens present in an individual’s sera sample. This enables simultaneous measurement of monoclonal antibody or sera cross-reactivity against multiple antigens in the same small sample. This method improves sensitivity, specificity, and limits measurement errors there might be in the single dimensional assays [59]. Microarray assays are another commonly used class of multidimensional assay that also measure antibody binding for multiple antigens simultaneously with one sample [60]. One difference between the mPlex multiplex assay and the microarray assay is that instead of binding antigens to different beads, the proteins are printed onto a glass slide for analyses [61]. The mPlex-Flu assay allows an ample amount of data to be collected at once, such as IgG, IgM, and IgA antibodies [59]. Using the mPlex-Flu allows for studies of how specific antibodies respond to the different viral strains, for example, H1 and H3 influenza HA antigens, or evolved seasonal H1 variants. This showcases the cross-reactivity of the antibodies and the level of response, allowing for conclusions to be made on the similarities between the strains. Since there is a wide range of detection, more data could be gathered than possible from an assay such as the ELISA or HAI.

Functional assays for the impact of antibody cross-reactivity involve measuring inhibition of viral binding to host cell receptors (e.g., competitive inhibition assays) or functional infection (e.g., microneutralization assays). The relevance to other multidimensional antibody binding assays is that functional assays better assess whether specific antibody binding will impact infection. Ultimately, in vivo vaccination, boost, and viral challenge studies, in either animal models or human studies, are the most clinically relevant to answering the questions of whether cross-reactive antibodies generated against a prior viral strain will protect against infection from a new, emergent influenza strain. Both microneutralization and viral challenge assays are often used as confirmatory assays to cross-reactive binding measurements. In influenza immunity studies, for example, ELISA and multiplex approaches have shown that antibodies against the stalk portion of HA can be quite cross-reactive. It has not yet been proven that these antibodies protect against viral infection in humans, though protective effects have been seen in mouse models [62].

### 5.2. Epitope Mapping Across Variants

The highest level of resolution for assessing B cell imprinting is to delineate the shared and discordant B cell epitopes present across the antigens of interest. Ideally, epitopes would be cataloged by a combination of protein structural analysis to identify amino acids in close spacial proximity on the antigen surface, and accessibility confirmed by in vitro assays demonstrating monoclonal antibody binding. For B cell epitopes, this is technically complicated by the fact that such epitopes may be either linear (i.e., a continuous sequence of 4–15 amino acids), or non-linear (i.e., spatially adjacent in 3 dimensions, but from non-contiguous areas of the amino acid sequence of the antigen). However, this level of resolution is both becoming technically feasible over the last several years. Several databases exist describing linear and non-linear epitopes for influenza and SARS-CoV-2 now exist (e.g., IEDB, AntiJen, or the Influenza Sequence and Epitope Database) [63]. In addition, recent bioinformatics advances in predicting and analyzing protein tertiary structures, including dimeric and trimeric protein complexes like influenza HA and SARS-CoV-2, have greatly accelerated epitope identification, and now make possible mapping of genetic variants to immunologic relevant epitopes [64].

Currently, immunologic epitope mapping using a large array of epitope-specific monoclonal antibodies to precisely confirm antibody binding to predicted epitopes has lagged behind bioinformatics methods. This is an area that molecular immunology could learn from the field of HLA (human leukocyte antigen) tissue typing and solid-organ transplant cross-matching, which has defined, in great detail, the epitopes present on HLA molecules, with both sequence analysis and antibody binding methods [65]. The HLA literature continues to define epitope level groupings of spatially adjacent linear and non-linear groupings of solvent-accessible amino acids on HLA molecules, called eplets [66,67,68], validated by monoclonal antibody studies [69]. Such information has been used to predict antibody reactivity in solid organ transplantation by knowing donor HLA molecular typing and recipient anti-HLA antibody specificities, a process called a virtual cross-match [70]. This prediction allows shortening the time to clear an organ donor-recipient pair for an organ transplant and allows prediction of how the development of anti-HLA antibodies will influence the percentage of organ donors by antibody specificity and HLA eplets [71,72,73]. Thus, similarly identifying and characterizing eplets on viral surface proteins may provide a more accurate method of characterizing antibody mediated B cell immunity to viral pathogens, and aiding in vaccine design.

### 5.3. Describing Immunologic Similarity: Antigenic Distance, Cartography, and Landscapes

The concept of antigenic distance has been developed to describe how similar two antigens (e.g., HA strains) are to one another [74] using a dimension-reduced metric. For antigenic distance, each dimension is represented by an antibody binding to a viral protein as represented by a string of features, for example, binding of IgG from a naive subject vaccinated and boosted with a single HA vaccine. The “antigenic distance” between any two HA’s is calculated using a Euclidean or another system (e.g., correlation distance, etc.), and is specific to the panel of antisera. A smaller antigenic distance may imply that more antibodies bind to both of the antigens [75]. A small antigenic distance could occur when antigens have many shared epitopes, or when high concentrations of a few antibodies are directed against a small number of epitopes. In 2004 the analytic method of antigenic cartography was developed as a way to map the evolution of a virus, specifically, the Smith group studied the H3N2 influenza virus and the variants that came from 1968 to 2003 and asked how immunologically close evolving strains were to each other. Antigenic distance was calculated using a panel of antisera generated by vaccinating and boosting ferrets against single strains of influenza. Different variants were clustered together using multi-dimensional scaling with antigenic distances calculated by antibody cross-reactivity from HAI assays. The clusters were then compared at the amino acid level to determine the sequence differences that were positively correlated with the antigenic distance [76].

Antigenic distance is currently used to determine which influenza strains should be included in seasonal vaccines [77]. When selecting a strain for vaccine development, it is preferred that the antigenic distance is low between the circulating strain and the strain used in the vaccine, allowing for the production of more protective antibodies to the circulating strain [74]. Additionally, it has been shown that birth year is a reliable predictor of the potential severity of seasonal influenza infection due to immune imprinting to strains encountered in childhood. When the antigenic distance between the seasonal strain and the childhood imprinted strain is smaller, infection rates decrease, indicating greater protection when seasonal strains are closely related to childhood imprinted stains [78]. Knowing the antigenic distance between strains may be useful for predicting the severity of seasonal influenza epidemics as well as identifying which groups will be most susceptible to the identified circulating strains.

Antigenic distances have previously been determined by HAI assay against a panel of influenza viruses and sera, building a titer matrix to calculate the antigenic distance. Additionally, antigenic distance has been quantified by comparing the sequences of the strains and which epitopes have been bound to antibodies [74]. The higher the sequence similarity between strains, the greater the amount of cross-reactive antibodies produced, leading to a more protective imprinted immune response [36,79]. Furthermore, as preexisting immunity plays a role in the immune response, it must also be taken into account along with antigenic distance when looking at antibody responses. This all plays a role in imprinting and looking at the similarity between strains could help indicate the cross-reactive and boosted immune response. The primed effect has been taken into account with a computer modeling system looking at antigenic-site specific antibodies and cross-reactivity and the computer results were similar to results from testing human sera with an ELISA looking at HA-specific antibody levels. This simulation showcases the idea that people with previous exposures to unique strains will also have different responses to vaccination or exposures to new strains [75].

Antibody landscapes are another way to visualize how pre-exposures will impact antibody protective responses. While related to antigenic distance, antibody landscape analysis looks at the intensity of antibody levels to generate an immune profile. The antigenic landscape allows for comparisons to be made looking at antigenic relationships, assessing how closely related strains are. This then provides information on cross-reactivity that occurs for different viral strains. Looking at antibody landscapes pre-vaccination and post-vaccination would lead to more conclusions for imprinting as it would exhibit the antibody levels for multiple strains allowing for easy comparison between past and more recent strains in circulation [14]. Being able to quantitate the distance between strains will aid in showing the strength of the potential effects of imprinting. Knowing how similar antigens are would allow for more research to be done looking at how conserved epitopes impact antibody responses.

### 5.4. Vaccination Strategies and Imprinting

Understanding the components of immune imprinting, specifically the competition between B-cell responses to epitope mutations and selective pressure on viral surface protein variants, has implications for vaccination strategy design. For example, multiple rounds of seasonal vaccination against the same influenza virus strain may result in a decrease in vaccine effectiveness, implying that current vaccination strategies may hinder immune responses to emerging influenza strains with a combination of conserved and novel epitopes, especially if the novel epitopes are located in the HA receptor binding site [24]. Other research has shown that repeated annual vaccination may lead to a decreased fold change in antibody responses post-vaccination, however, individuals receiving annual influenza vaccines have a higher protective baseline, though this study did not find any evidence of imprinting [80]. There is a wealth of evidence supporting the fact that exposure to a wild influenza virus generates a long-lasting immunological imprint with a long lasting protective effect against the assaulting strain and strains closely related to it. This response has been shown to be more significant than any imprinting response generated from influenza vaccination [81]. Although the vaccination response may not be as high in titer as the infection response, it has been shown in nonhuman primates that vaccinations provide a larger breadth of protection across seasons even with antigenic drift [23]. McCarthy et al. had also found that the recall response was greater in the non-human primate vaccination group than the infection group [23]. Overall, the effect imprinting has in conjunction with vaccinations is mixed and should be taken into consideration with the development of vaccines.

## 6. Immune Imprinting and SARS-CoV-2

### 6.1. The Challenge of Emerging SARS-CoV-2 Variants

Recently, the SARS-CoV-2 pandemic has garnered a great deal of attention from researchers trying to understand how the immune system responds to this new viral threat. The concept of imprinting, originally developed through the study of influenza viruses, is now being investigated in relation to SARS-CoV-2 [82]. First, it is important to understand that, like influenza, human coronaviruses are endemic, with SARS-CoV-2 being the most recent addition. There are several different coronaviruses circulating in the human population, with two main subgroupings, *α*− and *β*−, that cause upper respiratory illnesses [83]. SARS-CoV-2 falls into the *β*− category along with similarly highly pathogenic SARS-CoV-1 and Middle Eastern Respiratory Syndrome (MERS) viruses, as well as the less pathogenic OC43 and HKU1 variants [84]. Several serology studies have found that previous exposure to *β*− coronavirus variants can elicit antibodies that are capable of cross-reacting with SARS-CoV-2 antigens [85]. Although these antibodies may provide some protection against SARS-CoV-2 infection, their production may also impair the *de novo* development of SARS-CoV-2 specific neutralizing antibodies [86,87,88].

The concept of imprinting is also applicable to the antibody responses elicited against variants of SARS-CoV-2, where the response to the first variant encountered will likely be greater and more specific than subsequent encounters with divergent variants [89]. In addition, individuals will likely also demonstrate a higher response to any variant that may be more antigenically similar to the variants responsible for prior infections [90]. In support of this concept, it has been shown that individuals infected with the SARS-CoV-2 Delta variant also had a high IgG immune response to Kappa and Epsilon variants, which all share the L452R mutation. Additionally, the antibody breadth of protection against SARS-CoV-2 variants was higher in vaccinated individuals when compared to infected individuals [91]. However, two recent preprints suggest that vaccination against newer SARS-CoV-2 spike protein variants reengaged B cells and elicited responses to novel epitopes specific to such variants [92], and that SARS-CoV-2 receptor binding domain (RBD)-specific antibody responses six months after Omicron BA.1 breakthrough infection in mRNA-vaccinated individuals can result in new IgG responses to novel Omicron BA.1 epitopes [10]. Thus, while imprinting may hinder robust responses to new epitopes in viral variants, it is not absolute.

### 6.2. Evidence of SARS-CoV-2 Memory B Cell Imprinting

Studies of the proportion of B cells circulating specific to the receptor binding domain (RBD) of SARS-CoV-2 have found that people who were newly infected with SARS-CoV-2 had a greater amount of RBD-specific MBCs in their blood compared to the naive group [4], demonstrating a new SARS-CoV-2 specific “imprint” on the immune system. Additionally, newly infected individuals generated more SARS-CoV-2 RBD-specific memory B cells and neutralizing antibodies after a single vaccination as compared to previously naive individuals receiving a similar vaccination [4]. Furthermore, it has been demonstrated that individuals with repeated antigenic exposure, such as those receiving a third mRNA vaccine dose, exhibited a significantly greater number of RBD-binding memory cells than individuals receiving only two doses of the vaccine or even those who had a single previous SARS-CoV-2 infection [5]. This is indicative of memory B cell clonal expansion associated with repeat antigenic exposure. Importantly, unique MBC populations that were previously undetected grew in size during subsequent exposures with the same antigen. Such repeated exposures may lead to greater antibody cross-reactivity, potentially resulting in a greater breadth of neutralizing protection, better able to target newly emerging variants of the virus [5].

Memory B cells newly encountering SARS-CoV-2 spike protein epitopes may also induce cross-reactivity against similar *β*-HCoV spike proteins. Individuals vaccinated against the original Wuhan strain spike protein had IgG antibody levels similar to those of patients that exhibited severe symptoms from an infection [91]. When looking at the immune response against other *β*-coronaviruses, SARS-CoV-2 infected individuals showed a greater back-boosted effect, meaning they produced more antibodies against OC43 and HKU1 compared to those that received a vaccination. Upon the administration of subsequent vaccines, the vaccination group began to show an increase in antibody levels for OC43 and HKU1 as well [91]. McNaughton et al. found that a greater antibody reaction to additional β-HCoV spike proteins, such as OC43 or HKU1, was related to more severe SARS-CoV-2 infections stemming from the back-boosting effect associated with imprinting [15,19,93]. Supplementarily, there was no increase in the response seen for the *α*-HCoVs, such as NL63 or 229E [93].

### 6.3. Antibody Dependent Enhancement (ADE) vs. Imprinting

One concern regarding the development and administration of SARS-CoV-2 vaccines is the potential to induce antibody dependent enhancement (ADE) of infection [94,95,96,97]. ADE facilitates viral entry into cells, thereby enhancing rather than preventing infection, and this phenomenon has been reported with dengue virus infection and vaccination [98,99], as well as with cross-reactivity between Dengue and Zika viruses [100]. ADE, as it specifically relates to dengue virus, results from the production of neutralizing antibodies after infection with one serotype of the virus. However, upon secondary infection with a different serotype, preexisting antibodies directed against the first serotype are unable to fully neutralize the second. Rather, the sub or non-neutralizing antibodies bind to IgG Fc receptors on immune cells, facilitating cellular entry [100,101]. Vaccine associated disease enhancement has also been observed [102]. Theoretically, SARS-CoV-2 specific non-neutralizing antibodies may enhance host cell entry entry of the SARS-CoV-2 virus [103]. Indeed, it has been shown in vitro that coronavirus RBD-specific neutralizing monoclonal antibodies mediate the ADE of both Middle East respiratory syndrome (MERS-CoV) and SARS-CoV-1 coronaviruses by functionally mimicking viral receptors, which results in conformational changes in the RBD, allowing viral entry into host cells [104] Fortunately, to date, ADE appears to be more of a theoretical than significant clinical problem for SARS-CoV-2 infection and/or vaccination.

### 6.4. Relevance of Imprinting to SARS-CoV-2 Vaccine Development

Research into coronavirus imprinting may lead to the development of new vaccines and vaccination strategies. Indeed, for influenza vaccines, imprinting has been suggested to have both a negative and a positive impact on an individual’s immune response [23,24]. Preexisting antibodies have been shown to be less likely to neutralize new variants that arise as a result of antigenic drift [82]. The highly immunogenic globular head region of the influenza HA protein tends to be immunodominant, while the more conserved stalk region, is immunosubdominant [8]. Similarly, for SARS-CoV-2, a focus on the smaller conserved regions of the spike protein may be advantageous, limiting the negative effects of imprinting. Specifically, focusing on the RBD region of the spike protein, as it is immunosubdominant, may lead to a higher immune response with this more conserved region, potentially resulting in increased cross-reactivity between sub-variants that may arise [82]. Antigenic distance is also relevant to vaccination strategies as it may be used to determine when a new variant is unique enough to justify the development and distribution of a new vaccine. For example, the antigenic distance between the Delta variant and preceding variants was small enough that a new vaccine was not necessary [105], however, the antigenic distance between Omicron and other variants is large enough for researchers to consider developing a new vaccine formulation to produce new, variant-specific antibodies [106].

This issue may already be playing out with the SARS-CoV-2 bivalent vaccine [107] with the bivalent vaccines produced by Pfizer-BNT and Moderna. The first bivalent boosters contained mRNA designed to elicit immunity against the original WA1/2020 SARS-CoV-2 strain, present in the previous monovalent boosters, as well as the then newly emergent BA.1 strain. The results of these were disappointing, with only modest increases in anti-BA.1 neutralizing antibodies. As BA.1 was no longer circulating in the United States, the United States Food and Drug Administration approved new bivalent boosters directed against the, now dominant circulating variants BA.4 and BA.5. Results emerging from very recent studies suggest limited boosts in antibody levels with modest protection against target strains, with minimial increases in BA.4 and BA.5 protection from the WA1/2020 and BA.1 boosters [107,108,109]. These results are thought to be due to immune imprinting from multiple rounds of the prior WU1/2020 monovalent vaccine series [107]. In the future, it will likely be import to balance our newfound ability to rapidly create mRNA vaccines against emerging SARS-CoV-2 strains against the possibility that imprinting may blunt or negate our ability to protect against future variants.

## 7. Conclusions

Immune imprinting has developed as a concept from OAS and to a broader framework now useful for understanding the secondary immune response, as well as vaccine development based on preexisting immunity. This phenomenon is both beneficial and detrimental to responses against antigens containing previously encountered epitopes. Although the effect of imprinting may lead to a more robust response, it may also serve to hinder a more specific immune response or aid in viral infectivity. The impact of imprinting may be predicted by quantifying antigenic distance between strains of interest which may allow for insights into potential vaccine strategies. Immune imprinting can contribute to the understanding of other viral systems, such as SARS-CoV-2, to make predictions about viral circulation which may lead to the development of more effective action plans.

## Figures and Tables

**Figure 1 pathogens-12-00169-f001:**
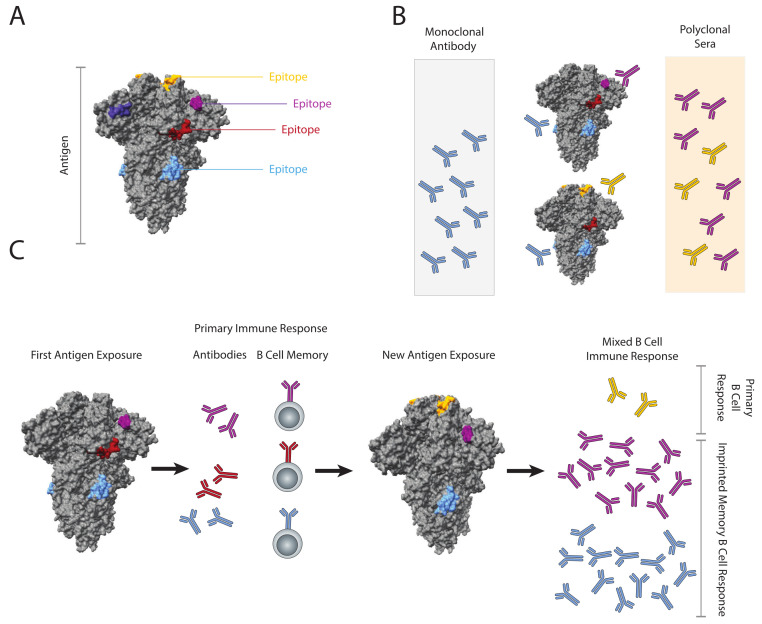
Antigens, Epitopes, and Imprinting of viruses. (**A**) Antigens contain many B cell epitopes, generally 3–15 amino acids and glycans clustered together on the protein surface that can be recognized by B cell surface immunoglobulin (i.e., B cell receptors) and trigger B cell immune responses. An example using the SARS-CoV-2 spike protein trimer with several epitopes, without glycans, is highlighted. (**B**) Antibody cross-reactivity occurs when antibodies bind to an epitope present on two different variants of the same protein, or a mixture of antibodies bind to different epitopes present on each variant. (**C**) Immune imprinting and B-cell memory. Primary exposure to a viral surface antigen leads to the production of antibodies against antigenic epitopes, and memory B cells. Exposure to a related antigen leads to a mixed primary and recall “imprinted” immune response. The primary response is less efficient and is overwhelmed by the rapid and robust memory B cell recall response to the imprinted antigens. The resulting response produces high levels of IgG to the imprinted epitopes and much lower levels of IgG against the new epitopes.

## Data Availability

Not applicable.

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
