# Peer review of "First Impressions Matter: Immune Imprinting and Antibody Cross-Reactivity in Influenza and SARS-CoV-2"

_pathogens, 2023, doi:10.3390/pathogens12020169_

Round 1

Reviewer 1 Report

This is an exceptionally well written history and review that brings us up to date on what used to be called original antigenic sin. It is highly relevant in influenza, it might be or is proposed to be relevant in SARS-CoV-2 (yes, there are some papers published on this), but there are also examples where evidence for this "imprinting" was not found or was a hypothesis rejected over the years in the vast immunology literature. The authors here emphasize positive evidence, and that works well, even as they do imply that one does not expect this "imprinting" as a universal concept that would apply to all immune responses. I am in favor of the paper as the authors have crafted it.  I have a few minor points of critique or suggested edits:

1) Line 107 states that there has not been evidence that naive and memory B cells would directly interfere with each other (reference 13). But later, line 194, it is explained how it is thought that this indeed occurs as they compete in germinal centers (reference 34). This sounds like a contradiction, but I believe if the authors could suggest a slight re-phrasing then one could see how both ideas expressed may be compatible rather than contradictory.

2) line 213, there is a possessive word missing an apostrophe.

3) line 300, "mice model" could be edited to "mouse model".

4) Section 6.1 title is about evidence for imprinting, but I find that section is about B cell memory, which though required for imprinting, is not sufficient to generate imprinting. There are examples of B cell memory and response to vaccination or infection without imprinting. I feel this subsection presents evidence for B cell memory in SARS-CoV-2 immune responses. Then in the next sub-section, we see reference to SARS-CoV-2 publications that do seem to implicate imprinting directly. If authors could introduce slight re-phrasing or re-titling, this minor issue could be resolved.

5) Line 512, "evolved" is meant in a conceptual sense, not genetic or molecular, and thus the intent might be disambiguated if that word were replaced with "developed as a theme" or something similar. The authors are good writers and I have confidence they could edit this well.

Author Response

1) Line 107 states that there has not been evidence that naive and memory B cells would directly interfere with each other (reference 13). But later, line 194, it is explained how it is thought that this indeed occurs as they compete in germinal centers (reference 34). This sounds like a contradiction, but I believe if the authors could suggest a slight re-phrasing then one could see how both ideas expressed may be compatible rather than contradictory.

Thank you for this observation.  We have revised this line to make it clear that that these ideas are compatible and complementary.

2) line 213, there is a possessive word missing an apostrophe.

We have corrected this.

3) line 300, "mice model" could be edited to "mouse model".

We have corrected this.

4) Section 6.1 title is about evidence for imprinting, but I find that section is about B cell memory, which though required for imprinting, is not sufficient to generate imprinting. There are examples of B cell memory and response to vaccination or infection without imprinting. I feel this subsection presents evidence for B cell memory in SARS-CoV-2 immune responses. Then in the next sub-section, we see reference to SARS-CoV-2 publications that do seem to implicate imprinting directly. If authors could introduce slight re-phrasing or re-titling, this minor issue could be resolved.

We thank the reviewer for this helpful comment, and have corrected this issue by a slight re-arrangement of the paragraphs, so that the section begins with antibody imprinting, and progresses to discuss the evidence for memory B cell imprinting.

5) Line 512, "evolved" is meant in a conceptual sense, not genetic or molecular, and thus the intent might be disambiguated if that word were replaced with "developed as a theme" or something similar. The authors are good writers and I have confidence they could edit this well.

We have edited this sentence to include the phrase “developed as a theme”.

Reviewer 2 Report

This manuscript reviewed knowledge of imprinting for influenza and for emerging COVID-19. 

Imprinting with respect to influenza has been widely covered in the literature, whilst the concept is of interest for antibody responses to SARS-CoV-2. Thus the manuscript would benefit from some editing in the sections covering influenza. 

How imprinting could/should be measured and how vaccines could/should be tailored to minimse this observation would be of interest.

Other minor comments:

Line 32 – add abbreviation for MBC

Figure 1A – glycans are mentioned in the legend but there are no glycans in the figures

Figure 1C – the primary antibody response generated 2 x Abs to each antigen, whilst the new antigen exposure also generates 2 x Abs to the yellow antigen encountered for the first time. In regard to the understanding of imprinting, whether these responses are supposed to be equivalent, or slightly reduced is of interest, and perhaps should be addressed, or acknowledged?

Lines 155-171 – influenza viruses are grouped into influenza A subtypes (eg H1N1, H3N2) and influenza B lineages (B/Victoria, B/Yamagata). Within these subtypes or lineages, there are individual influenza virus strains. Please adjust the language for lines 155-171 as follows:

  “In addition to different strains subtypes, there is also antigenic drift within the same strain subtype. Marchi et al. focused on three of the past H1N1 strains recommended used for inclusion in influenza vaccines vaccinations, A/Brisbane/59/2007 which was used for the 2008/2009 and 2009/2010 seasons, A/California/07/2009 which was used for the flu seasons from 2010/2011 to 2016/2017, and A/Michigan/45/2015 which was used for the seasons 2017/2018 and 2018/2019 [28]. It shows that there are different levels of cross-reactivity and protection when looking at immune responses to these strains within the H1 subtype as measured by hemagglutination inhibition (HAI)and single radial hemolysis assays using human serum samples [28]. These high serum IgG binding cross-reactivity between the A/California/07/2009 and the A/Michigan/45/2015 H1N1 even when the California 2009 first began to circulate. Additionally, within the elderly adult population, there was a greater number of cross-reactive and cross-protective antibodies seen with all three strains likely due to their primed and repeated exposures to subtypes H1N1 strains that circulated earlier in their lives.[28]. This shows that imprinting does not just have to do with different strains or groups subtypes or lineages of the viruses, but can also have an impact within the same strain subtype that has experienced antigenic drift. Thus one of the benefits of imprinting has to do with the concept that the immune system can quickly respond to different, yet related viruses or multiple strains of the same virus.

Line 404 – please clarify the term use of the word ‘wild’? i.e. “to a wild influenza virus”

Author Response

Reviewer 2

This manuscript reviewed knowledge of imprinting for influenza and for emerging COVID-19. 

Imprinting with respect to influenza has been widely covered in the literature, whilst the concept is of interest for antibody responses to SARS-CoV-2. Thus the manuscript would benefit from some editing in the sections covering influenza. 

We have edited and shortened some sections to shift the emphasis to SARS-CoV-2.  Because the history and conceptual development of is heavily influenza based, we do need to include a moderate amount of this information for review.

How imprinting could/should be measured and how vaccines could/should be tailored to minimse this observation would be of interest.

We have added a short explanation, trying to balance the addition with Reviewer 3’s comments.

Other minor comments:

Line 32 – add abbreviation for MBC

This has been added at Line 33.

Figure 1A – glycans are mentioned in the legend but there are no glycans in the figures

The manuscript section has been corrected

Figure 1C – the primary antibody response generated 2 x Abs to each antigen, whilst the new antigen exposure also generates 2 x Abs to the yellow antigen encountered for the first time. In regard to the understanding of imprinting, whether these responses are supposed to be equivalent, or slightly reduced is of interest, and perhaps should be addressed, or acknowledged?

Respectfully, we have already addressed this in the explanation, and the figure illustration shows very different primary and secondary The second response to antigens in Figure 1C (right figure) shows a modest primary response (schematic of 2 yellow IgG)and a large secondary response (2 IgG -> 11 IgG purple and 2 IgG -> 12 IgG Blue).  These responses are not supposed to be equivalent, as explained in the caption “Exposure to a related antigen leads to a mixed primary and recall "imprinted" immune response. The primary response is less efficient and is overwhelmed by the rapid and robust memory  B cell recall response to the imprinted antigens. The resulting response produces high levels of IgG to the imprinted epitopes and much lower levels of IgG against the new epitopes.”

Lines 155-171 – influenza viruses are grouped into influenza A subtypes (eg H1N1, H3N2) and influenza B lineages (B/Victoria, B/Yamagata). Within these subtypes or lineages, there are individual influenza virus strains. Please adjust the language for lines 155-171 as follows:

  “In addition to different strains subtypes, there is also antigenic drift within the same strain subtype. Marchi et al. focused on three of the past H1N1 strains recommended used for inclusion in influenza vaccines vaccinations, A/Brisbane/59/2007 which was used for the 2008/2009 and 2009/2010 seasons, A/California/07/2009 which was used for the flu seasons from 2010/2011 to 2016/2017, and A/Michigan/45/2015 which was used for the seasons 2017/2018 and 2018/2019 [28]. It shows that there are different levels of cross-reactivity and protection when looking at immune responses to these strains within the H1 subtype as measured by hemagglutination inhibition (HAI)and single radial hemolysis assays using human serum samples [28]. These high serum IgG binding cross-reactivity between the A/California/07/2009 and the A/Michigan/45/2015 H1N1 even when the California 2009 first began to circulate. Additionally, within the elderly adult population, there was a greater number of cross-reactive and cross-protective antibodies seen with all three strains likely due to their primed and repeated exposures to subtypes H1N1 strains that circulated earlier in their lives.[28]. This shows that imprinting does not just have to do with different strains or groups subtypes or lineages of the viruses, but can also have an impact within the same strain subtype that has experienced antigenic drift. Thus one of the benefits of imprinting has to do with the concept that the immune system can quickly respond to different, yet related viruses or multiple strains of the same virus.

The requested changes have been made.

Reviewer 3 Report

The present manuscript by Samantha et al. entitled “First Impressions Matter: Immune Imprinting and Antibody Cross-reactivity in Influenza and SARS-CoV-2” approach the well-known evidence of the original antigenic sin in influenza and current evidence suggesting the OAS occur in SARS-COV-2 infections. The authors also include methods to study the OAS and its implications in vaccine design.

Minor improvements needed:

1.     The manuscript is too extensive. Some sections should be shorter.

2.      Lines 18-46. The authors need to include some references.

3.      In my opinion, “4. The role of memory B cells in imprinting” and “5. Measuring imprinting and antibody cross-reactivity” should be before “3. Immune imprinting and influenza”.

4.      A phylogenetic tree of influenza may be useful to describe the types of influenza viruses.

Author Response

  1. The manuscript is too extensive. Some sections should be shorter.

We have edited the manuscript in general to make it more concise.

  1. Lines 18-46. The authors need to include some references.

Additional references have been added.

  1. In my opinion, “4. The role of memory B cells in imprinting” and “5. Measuring imprinting and antibody cross-reactivity” should be before “3. Immune imprinting and influenza”.

We spent considerable time trying to re-work the manuscript to accommodate this opinion.  Unfortunately, this led to a lack of clarity and confusion when explain the concept of imprinting.  For example, discussing antigens and epitopes in general, and memory B cell responses, before discussing the concepts of influenza evolution before discussing the historical development based on observations with influenza immunity was, in our hands, not workable.  We have, therefore, left the ordering of sections as originally conceived.

  1. A phylogenetic tree of influenza may be useful to describe the types of influenza viruses.

In the interest of brevity, as requested in comment 1, and balancing the request of Reviewer 1 to focus more on SARS-CoV-2, we have not expanded the manuscript with a phylogenetic tree of influenza strains.